# Characterization of Defensin-like Protein 1 for Its Anti-Biofilm and Anti-Virulence Properties for the Development of Novel Antifungal Drug against *Candida auris*

**DOI:** 10.3390/jof8121298

**Published:** 2022-12-14

**Authors:** Majid Rasool Kamli, Jamal S. M. Sabir, Maqsood Ahmad Malik, Aijaz Ahmad

**Affiliations:** 1Department of Biological Sciences, Faculty of Science, King Abdulaziz University, Jeddah 21589, Saudi Arabia; 2Center of Excellence in Bionanoscience Research, King Abdulaziz University, Jeddah 21589, Saudi Arabia; 3Department of Chemistry, Faculty of Science, King Abdulaziz University, Jeddah 21589, Saudi Arabia; 4Department of Clinical Microbiology and Infectious Diseases, School of Pathology, Faculty of Health Sciences, University of the Witwatersrand, Johannesburg 2193, South Africa; 5Division of Infection Control, Charlotte Maxeke Johannesburg Academic Hospital, National Health Laboratory Service, Johannesburg 2193, South Africa

**Keywords:** *C. auris*, D-lp1, anti-biofilm, anti-virulence, membrane integrity

## Abstract

*Candida auris* has emerged as a pan-resistant pathogenic yeast among immunocompromised patients worldwide. As this pathogen is involved in biofilm-associated infections with serious medical manifestations due to the collective expression of pathogenic attributes and factors associated with drug resistance, successful treatment becomes a major concern. In the present study, we investigated the candidicidal activity of a plant defensin peptide named defensin-like protein 1 (D-lp1) against twenty-five clinical strains of *C. auris*. Furthermore, following the standard protocols, the D-lp1 was analyzed for its anti-biofilm and anti-virulence properties. The impact of these peptides on membrane integrity was also evaluated. For cytotoxicity determination, a hemolytic assay was conducted using horse blood. The minimum inhibitory concentration (MIC) and minimum fungicidal concentration (MFC) values ranged from 0.047–0.78 mg/mL and 0.095–1.56 mg/mL, respectively. D-lp1 at sub-inhibitory concentrations potentially abrogated both biofilm formation and 24-h mature biofilms. Similarly, the peptide severely impacted virulence attributes in the clinical strain of *C. auris*. For the insight mechanism, D-lp1 displayed a strong impact on the cell membrane integrity of the test pathogen. It is important to note that D-lp1 at sub-inhibitory concentrations displayed minimal hemolytic activity against horse blood cells. Therefore, it is highly useful to correlate the anti-*Candida* property of D-lp1 along with anti-biofilm and anti-virulent properties against *C. auris*, with the aim of discovering an alternative strategy for combating serious biofilm-associated infections caused by *C. auris*.

## 1. Introduction

Advancements in the field of surgery and medicine have been a boon to human society. Still, on the other hand, they have built up a population of immunocompromised individuals facing a high risk of opportunistic infections in their daily lives [1]. These patients are prone to opportunistic fungal infections, mostly candidiasis caused by pathogenic *Candida* species. *Candida auris* is an emerging pathogen resulting in deep-seated bloodstream infections in people suffering from other immunocompromised situations. Presently, this pathogenic yeast is labeled as a multidrug-resistant hospital-born fungal pathogen and is evolving as a menace to public healthcare units worldwide [2]. Contrary to other *Candida* species, this pathogenic yeast bears the potential to withstand tough environments outside the host, which advocates the strength of pathogenic attributes present in *C. auris*. Owing to its ability to cause outbreaks, misidentification, and tolerance to commonly used drugs, *C. auris* is a disaster for patients who are already in health compromising situations [3].

Plant defensins have a conserved structure that is cysteine-rich, and thus, distinguished as a cysteine-stabilized αβ-motif [4,5,6]. Antimicrobial properties of plant defensins against a broad range of microbes have been well reported in the literature [7,8,9,10,11] whereas, in general, these peptides are non-toxic in humans [12,13,14]. Upon exploring the mechanism of action, plant defensins have been reported to disrupt fungal membranes and thereby either move inside the cell and affect the functionality of vital proteins or remain confined to the fungal outer envelops [15,16,17]. There are various antifungal modes of action of plant defensins, but most reported are cellular apoptosis and the generation of reactive oxygen species (ROS) [18]. Furthermore, there have been reports on anti-biofilm activity of plant defensins against *C. albicans* [11]. As biofilms are tolerant to many conventional therapies and only some novel agents, liposomal-amphotericin B, caspofungin, miconazole, and anidulafungin can be utilized for this purpose [19,20,21]. Various fungal species are known to form biofilms. However, *Candida* species is one of the most predominant fungi that play an important essential role in single- and mixed-species biofilm formation, and is also responsible for several biofilm-associated infections in immunocompromised patients [22]. Therefore, identifying a novel antibiofilm compound is a pressing need.

Similarly, D-lp1 (formerly known as gamma-hordothionin) is a sulphur-rich defensin present in barley endosperm, with a molecular weight mass of 5.25 kDa and a sequence length of 47 amino acids, and has been reported to show inhibitory activity against common food decaying yeast, *Zygosaccharomyces bailii, Zygosaccharomyces rouxii, Kluyveromyces lactis, Saccharomyces cerevisiae*, and *Debaryomyces hansenii* [23]. However, the anti-*Candida* effect of D-lp1 has not been explored by researchers; therefore, this study focuses on exploring the antifungal, anti-biofilm, and anti-virulent properties of D-lp1 against drug-resistant *C. auris*.

## 2. Materials and Methods

### 2.1. Candida auris Strains and Peptide

In this study, twenty-five clinical strains of *Candida auris* were used. All the strains were procured from the National Institute of Communicable Diseases (NICD), Johannesburg, South Africa (Table 1). *Candida albicans* ATCC90028 and *Candida parapsilosis* ATCC 22019 stored in the department were used as reference and quality control strains, respectively. The Human Research Ethics Committee of the University of the Witwatersrand granted an ethics waiver (M140159) to use *C. auris* MRL6057 for research purposes. Furthermore, the strains were stored at −80 °C in the form of glycerol stocks, and were revived and maintained on yeast extract peptone dextrose (YPD) agar plates for experimental use.

Defensin-like protein 1 was purchased from Thermo Fisher Scientific, IL, USA, with a purity of ≥80%. The peptide purchased was customized with a peptide sequence of RICRRRSAGFKGPCVSNKNCAQVCMQEGWGGGNCDGPLRRCKCMRRC. As per the supplier’s certificate, the purity of the peptide was determined by a high-performance liquid chromatography (HPLC) system (Biosynthesis, Inc., Lewisville, TX, USA). The stock solution of 64 mg/mL of the D-lp1 was prepared using 1% DMSO, and was kept in a dark environment at −20 °C until use.

### 2.2. Antifungal Susceptibility Profiling

Minimum inhibitory concentrations (MIC) of D-lp1 and amphotericin B (AmB; Sigma-Aldrich, St. Louis, MO, USA) against all the strains of *C. auris* were evaluated by the broth microdilution method following CLSI guidelines [24]. For reference and quality control, *C. albicans* ATCC90028 and *C. parapsilosis* ATCC 22019 were included in each set of experiments. Briefly, the stock solution of D-lp1 and AmB was prepared in 1% dimethyl sulfoxide (DMSO; Merck, RSA) and test concentrations used for the experiment ranged from 16 × 10^−3^–8 × 10^−6^ mg/mL and 25–0.02 mg/mL for AmB and D-lp1, respectively. The plates were incubated at 35 °C for 24 h. Additionally, the experiment included sterility as well as growth control. Later, MIC was determined by visual observation and recorded as the lowest concentration of D-lp1 that inhibited the growth of *C. auris* compared to the growth controls.

After that, minimum fungicidal concentration (MFC) was also evaluated. For this purpose, aliquots 10 μL samples from each well above the MIC were withdrawn and sub-cultured on Sabouraud dextrose agar (SDA; Merck, Darmstadt, Germany, RSA) growth media and incubated for 24 h at 35 °C. Furthermore, MFC was recorded as the lowest concentration that inhibited *C. auris* growth on agar plate [25].

### 2.3. Investigating the Anti-Biofilm Property of D-lp1 in C. auris

The impact of D-lp1 on the adherence of *C. auris*, the inhibition of biofilm formation, and the disruption of mature biofilm was evaluated as previously described [26]. The *C. auris* strains from Group-C, showing the highest MIC and MFC values, were undertaken for biofilm assay.

Briefly, for the biofilm assay, 2 mL of freshly prepared inoculum (0.5 McFarland) was seeded in 12-well polystyrene plates. The plates were incubated at 37 °C for 90 min, allowing the cells to adhere for biofilm formation. Thereafter, the wells were gently washed with PBS, a sterile growth medium was added to each well, and plates were incubated at 37 °C for 24 h. For investigating the inhibitory effect of plant-defensin over the adherence of *C. auris,* the freshly prepared inoculum was seeded in 12-well polystyrene plates along with the test agent (4 × MIC and lower concentrations) and incubated at 37 °C for 90 min (adherence phase of biofilm formation). To study the inhibitory role of D-lp1 on biofilm formation, the *C. auris* cells were allowed to adhere to the polystyrene plate. Later, the biofilm was treated with various concentrations of the test agent for 24 h at 37 °C.

To investigate the disruptive nature of the test plant defensin on mature biofilms, the *C. auris* biofilms were grown for 24 h, the medium was aspirated from each well followed by adding fresh medium supplemented with different concentrations of D-lp1, and incubated for an extra 24 h. The experiment was conducted in triplicate at least twice, and growth and sterility controls were included. Additionally, to evaluate the fungal viability within the biofilms, XTT reduction assay was performed as described elsewhere [27]. The percentage inhibition of biofilm was calculated using the following equation.
(1)$$\%\;\mathrm{Biofilm}\;\mathrm{inhibition}=\frac{\mathrm{Control}\;\mathrm{OD}490\;\mathrm{nm}-\mathrm{Test}\;\mathrm{OD}490\;\mathrm{nm}}{\mathrm{Control}\;\mathrm{OD}490\;\mathrm{nm}}\times 100$$

### 2.4. Scanning Electron Microscopy (SEM) of Mature Biofilm Formed by C. auris MRL6057

The anti-biofilm property of D-lp1 against the most stubborn strain of *C. auris*, MRL6057, was further evaluated with the help of SEM by following the method previously described [28]. Briefly, the *C. auris*, MRL6057 cells were grown on glass coverslips under standard biofilm growing conditions for 24 h, the medium was aspirated, gently washed with PBS followed by adding fresh medium supplemented with D-lp1 (4 × MIC), and incubated for an extra 24 h. After incubation, the culture broth was removed, and sessile cells were given a gentle wash with PBS followed by fixing with glutaraldehyde. Then, the biofilms were carefully washed and gradually dehydrated with ethanol. Later, the coverslips bearing fixed and dehydrated biofilms were subjected to critical point drying, carbon-coated, and observed under the SEM (Zeiss Gemini 2 Crossbeam 540 FEG SEM). The experiment was conducted in triplicate at least twice; negative controls were included in the study.

### 2.5. Investigating the Effect of D-lp1 on Cellular Viability of C. auris

D-lp1 was examined for its fungicidal activity against *C. auris* MRL6057 using Muse^TM^ Count, and the company provided a Viability kit and the method used. Briefly, cells at a concentration of 5.0 × 10^6^ CFU/mL (0.5 McFarland) were treated with various concentrations of D-lp1 (0.5 × MIC, MIC, and MFC), then were washed (PBS) and mixed with the provided reagents in an appropriated proportion. The samples were incubated for 5 min at room temperature away from light and examined using a Muse^TM^ cell analyzer machine. Cells treated with 10 mM H_2_O_2_ (Merck, Germany) and healthy growing cells were used as positive and negative controls, respectively.

### 2.6. Investigating the Effect of D-lp1 on Membrane Integrity

The membrane-disrupting ability of *C. auris* MRL6057 was scrutinized by Propidium Iodide (Sigma-Aldrich) dye, as it is a commonly used agent for analyzing the intactness of plasma membrane [29]. The experiment was adopted from elsewhere [30]. Briefly, cells were grown in standard conditions for 24 h, washed two times (PBS), resuspended in SDB at a concentration of 0.5 McFarland, and exposed to D-lp1 (0.5 × MIC, MIC, and 2 × MIC) for 4 h. Then, cells were washed and stained with PI (30 μM), and samples were observed with fluorescence microscopy (Zeiss Laser Scanning Confocal Microscope (LSM) 780 and Airyscan) (Carl Zeiss, Inc., Macquarie Park, NSW, Australia). Both positive and healthy controls were included in the study.

### 2.7. Investigating the Effect of D-lp1 on Secretory Aspartyl Proteinase (SAPs)

The effect of test plant defensin on the production of SAPs in *C. auris* MRL6057 was analyzed using the method previously described [31]. Briefly, the test strain was grown for 18 h at 37 °C followed by centrifugation (3000 rpm, 5 min) and resuspended in fresh growth media. Turbidity was adjusted to 0.5 McFarland and exposed to the desired concentrations of D-lp1 (0.25 × MIC, 0.5 × MIC, and MIC) for 4 h. Later, aliquots (2 μL) were placed at equidistant points on proteinase agar plates (agar, 2%; BSA 2 g; yeast nitrogen base w/o amino acids; ammonium sulphate 1.45 g; glucose 20 g; distilled water to 1 L). Then, plates were incubated for 3–4 days at 37 °C, and SAP activity was calculated in terms of Pz value (diameter of the colony divided by the diameter of the colony plus zone of clearance).

### 2.8. Investigating the Effect of D-lp1 on Adherence to Epithelial Cells

The impact of D-lp1 on the attachment of *C. auris* MRL6057 to the epithelial cells was analyzed using the method previously described [32]. Briefly, the *C. auris* MRL6057 (0.5 McFarland) were exposed to test peptides at various concentrations (0.25 × MIC, 0.5 × MIC, and MIC) for 4-h time periods. Afterward, the cells were centrifuged, washed twice with PBS, and the turbidity was adjusted to 0.5 McFarland. In a parallel experiment, the buccal epithelial cells were collected using a sterile swab, suspended in sterile PBS, washed, and re-suspended in 2 mL sterile PBS, and adjusted to a turbidity of 10^5^ CFU/mL using a hemocytometer. Later, aliquots of yeast cells and oral epithelial cells were mixed in equal ratios and incubated at 37 °C, 60 rpm for 2 h. After incubation, for separating non-adherent *C. auris* cells from epithelial cells, a 20 μm nylon filter was used, followed by washing the epithelial cells and re-suspending them in distilled water (1 mL). Later, slides were prepared, and gram staining was conducted. The amount of adherent *C. auris* cells per 100 epithelial cells was counted using a light microscope (Leica DM 500 microscope). The experiment included the untreated cells as a negative control.

### 2.9. Investigating the Effect of D-lp1 on Efflux Assay

To analyze the effect of D-lp1 on efflux assay, extracellular Rhodamine 6G (R6G) assay was used as described elsewhere [33]. Briefly, *C. auris* MRL6057 cells were subjected to 4-h exposure to D-lp1 (0.5 × MIC and MIC) followed by a de-energization step, where 2,4 dinitrophenol, 2-deoxy-D-glucose were added to cells for 45 min, and later, 10 μM of R6G was added and kept under 40-min incubation at room temperature. Then, cells were washed and resuspended in glucose-free PBS. Aliquots (1 mL) were withdrawn after 5-min intervals for 1 h, centrifuged (3000 rpm), and the supernatant was used to record OD at 527 nm. To investigate energy-dependent efflux, glucose (0.1 M) was added after 20 min of incubation, aliquots were taken at 5-min intervals, and the sample was centrifuged to collect the supernatant. To quantify the actual concentration of R6G being thrown out of the cells, the recorded OD_527_ was compared with the standard concentration curve of R6G.

### 2.10. Intracellular R6G Accumulation Assay

The intracellular accumulation of R6G was examined using the method previously described [33]. Briefly, *C. auris* MRL6057 was exposed, similarly to the above, resuspended in PBS supplemented with glucose (2%) and R6G (4 µM), and kept for 30 min at room temperature. After that, centrifugation was conducted, cells were mixed with PBS, and slides were made for analysis through fluorescence microscopy (Zeiss Laser Scanning Confocal Microscope (LSM) 780 and Airyscan) (Carl Zeiss, Inc.).

### 2.11. Cytotoxicity Analysis of D-lp1

D-lp1 was subjected to cytotoxicity analysis using horse blood (NHLS, Johannesburg, South Africa). The protocol was followed as described in the previously published work [34]. Briefly, horse blood (10 mL) was centrifuged (3000 rpm for 10 min) and the supernatant was discarded. The remaining pellet was rinsed three times and resuspended in chilled PBS at a final concentration of 10%, and was used as the stock solution. The stock solution was then diluted (1:10) and an aliquot (100 μL) was mixed with various concentrations of D-lp1 (0.25 × MIC, 0.5 × MIC, MIC, and MFC). Incubation was conducted for 1 h at room temperature, and later, tubes were centrifuged (2000 rpm for 10 min). The supernatant was used to record the absorbance at 450 nm using microplate readers (Molecular Devices, San Jose, CA, USA). The hemolysis percentage was estimated using the equation below, where triton X-100 (1%) was the positive control, and PBS was the negative control.
(2)$$\%\;\mathrm{Haemylosis}=\frac{\left[\left(A450\;\mathrm{of}\;\mathrm{treated}\;\mathrm{sample}\right)-\left(A450\;\mathrm{of}\;\mathrm{negative}\;\mathrm{control}\right)\right]}{\left[\left(A450\;\mathrm{of}\;\mathrm{positive}\right)-\left(A450\;\mathrm{of}\;\mathrm{negative}\;\mathrm{control}\right)\right]}\times 100$$

### 2.12. Statistical Analysis

All the experiments were carried out in triplicate at least two times, and the results were presented as average ± standard error of the mean. Statistical analysis was conducted in Graph Pad Prism version 9.1.0 utilizing Student’s unpaired two-tailed *t*-tests and two-way ANOVA. *p* value < 0.05 was considered statistically significant.

## 3. Results and Discussion 

### 3.1. Candidicidal Activity of D-lp1

All the clinical strains of *C. auris* were found to be highly susceptible to D-lp1, with the MIC values ranging from 0.047–0.78 mg/mL, whereas the MFC values ranged from 0.095–1.56 mg/mL. Based on the antifungal susceptibility results, *C. auris* strains were clustered into three groups (A–C), where group “C”, consisting of *C. auris* MRL2921, MRL4000, MRL5762, MRL5765, and MRL6057, was found to have the highest MIC and MFC values, 0.78 mg/mL and 1.56 mg/mL, respectively, among all tested strains of *C. auris* (Table 1). Additionally, the sterility control wells were clear and no interference of DMSO (1%) was observed in the growth control wells. As previously reported [25], group A and B were sensitive to AmB with MIC values ranging from 0.125 × 10^−3^–0.1 × 10^−3^ mg/mL. While *C. auris* strains belonging to group C were categorized as AmB resistant [35], *C. auris* MRL2921, MRL4000, MRL5762, and MRL5765 were found to have MIC values of 0.002 mg/mL, whereas MRL6057 was reported as the most resistant strain with an MIC value of 0.004 mg/mL. As expected, the MIC and MFC values for AmB against *C. albicans* ATCC90028 were recorded as 0.125 × 10^−3^ and 0.5 × 10^−3^ mg/mL, respectively, whereas these figures against *C. parapsilosis* ATCC 22019 were recorded as 0.25 × 10^−3^ and 0.5 × 10^−3^ mg/mL, respectively. The test peptide D-lp1 also showed potent antifungal activity against both the reference and quality control strains, with MIC and MFC values ranging from 0.047 to 0.376 mg/mL. Although the MIC and MFC values of the test peptides seem higher than AmB, being non-cytotoxic in nature, the D-lp1 has more promising antifungal effect against *C. auris*, which is a shortcoming of AmB. Furthermore, many researchers have demonstrated the low sensitivity of *C. auris* to AmB [36,37,38]. Thus, it is most important to note that the D-lp1 at a low concentration of 0.39 mg/mL and 1.56 mg/mL displayed candidicidal activity against these AmB-resistant strains of *C. auris*, and, therefore, has good potential in the field of antifungal drug development.

Plant defensins have emerged as attractive materials for developing novel antifungals, and, most importantly, they are usually non-toxic against human cells [12,13,14]. The present study agrees with previous findings, where plant defensins such as Lc-def [39], HsAFP1 [11,40], and ZmD32 [41] have been reported to display inhibitory activity against pathogenic *Candida* species. In a separate study, D-lp1 was found to be active against *Z. bailii* and *D. hansenii*, and the MFC ranged from 0.05–0.1 mg/mL, while the inhibitory concertation for *S. cerevisiae* and *Z. rouxii* was found to be 0.4 mg/mL [23]. However, in the current study, D-lp1 was found to be active against clinical strains of *C. auris* at varying concentration ranges. Therefore, it is worthwhile to investigate candidicidal activity, as well as its impact on various virulence attributes in *C. auris*.

### 3.2. D-lp1 Is Effective against C. auris Adhesion, Biofilm Formation, and Mature Biofilms

In a previous study [25], *C. auris* strains belonging to Group C were reported to be suitable biofilm formers. Therefore, they were subjected to the biofilm assay. For investigating the anti-biofilm property of the test plant defensin, three different biofilm assays, effects on cell adhesion, biofilm formation, and disruption of mature biofilms were conducted. In the adherence inhibition biofilm assay, D-lp1 at a low concentration, 0.19 mg/mL, altered the adherence of *C. auris* MRL2921, MRL4000, MRL5762, and MRL5765, whereas, at 0.39 mg/mL, the adherence of MRL6057 was inhibited. Compared to the average value of the three negative controls, there was ≥90% inhibition in the adherence of *C. auris* cells, which was confirmed by XTT assay (Figure 1A). Similarly, in the developmental inhibition, D-lp1 abrogated biofilm formation in all of the *C. auris* strains of group C. The results obtained from XTT assay advocated the anti-biofilm property of D-lp1 at a low concentration of 0.78 mg/mL (MIC) and 1.56 mg/mL (MFC or 2 × MIC). All clinical strains, except MRL6057, showed ≥ 90% inhibition in biofilm formation in MIC value, whereas the biofilm formation capability of MRL6057 was abrogated at 2 × MIC (Figure 1B). Furthermore, in the biofilm disruption assay, ≥ 90% disruption in mature biofilms formed by *C. auris* MRL2921, MRL4000, MRL5762, and MRL5765 was observed at a concentration of 1.56 mg/mL (MFC), whereas a higher concentration of 3.12 mg/mL (4 × MIC) was responsible for damaging mature biofilms formed by MRL6057 (Figure 1C). Altogether, the test plant defensin showed concentration-dependent inhibition of biofilm formation and mature biofilm relative to the untreated control. Additionally, among five tested clinical strains, *C. auris* MRL6057 was the best biofilm former with the highest metabolic activity. Furthermore, it countered the anti-biofilm nature of D-lp1, suggesting the abundance of active drug efflux pumps and the presence of other associated resistance attributes in the biofilms formed by *C. auris* MRL6057. However, D-lp1 was found consistent in inhibiting, as well as eradicating, the biofilm formed by *C. auris* MRL6057, which strongly advocates the potential anti-biofilm properties of the test peptide against clinical strains of *C. auris*.

The main point of concern is that biofilm-associated infections display a higher level of antifungal drug resistance, and only a few therapeutic agents can be employed to solve this problem [19,20,21]. Previous studies have proven the anti-biofilm activity of plant defensins, recombinant (r) HsAFP1, which prevented *C. albicans* biofilm formation. In contrast, it could not destroy the mature biofilms formed by *C. albicans* [11]. Similarly, a linear 19-mer peptide derived from HsAFP1 has shown strong in vitro anti-biofilm activity against *C. glabrata*, *C. albicans*, and caspofungin-resistant *C. albicans* strains; additionally, it reduced biofilm formation of *C. albicans* in vivo when combined with caspofungin [40]. Equally, the in vitro anti-biofilm activity of D-lp1 was proven in the present study, and its potency in in vivo models needs to be further investigated.

### 3.3. SEM Analysis Showed the Ability of D-lp1 to Disrupt Mature Biofilm

The disruptive ability of D-lp1 against mature biofilms formed by *C. auris* MRL6057 was monitored by SEM (Figure 2). A comparative evaluation of D-lp1-treated and untreated mature biofilm was conducted to examine the anti-biofilm property of the test peptide against *C. auris* MRL6057. As observed at lower magnification (20 µm), the coverslip with the untreated control consisted of a complex three-dimensional network of highly dense and scattered yeast cells (Figure 2A). Further investigation at higher magnification (2 µm) showed that the cells were healthy and uniform with smooth surface architecture, whereas treatment with D-lp1 at 3.12 mg/mL (4 × MIC) resulted in the abrogation of mature biofilm (Figure 2B). A closer view showed that cells were distorted and unhealthy, along with scarce extracellular matrix; additionally, some dead, ruptured, and degraded cells were also spotted when examined under SEM. The ultrastructural imaging further assured the results obtained from XTT assay, and pointed towards the cell membrane and cell-wall-damaging properties of the test peptide against *C. auris* MRL6057.

Based on these results, an in-depth study was carried out to evaluate the impact of D-lp1 on *C. auris* MRL6057 cell viability, membrane integrity, proteinase production, efflux pumps, and expression of genes associated with biofilm formation and drug efflux pumps.

### 3.4. D-lp1 Hinders Growth and Viability of C. auris MRL6057

The results obtained using the Muse^TM^ cell analyzer further confirmed the candidicidal activity of D-lp1 (Figure 3). The negative control displayed 100% cell viability, unlike in the positive control, where only 3.6% of cells were found live. Similarly, at sub-inhibitory concentrations, 0.39 mg/mL (0.5 × MIC), the percent viability was recorded as 50.4%, whereas a sharp decline in the percent viability was recorded at 0.78 mg/mL (MIC) and 1.56 mg/mL (MFC), which was 24.9% and 9.5%, respectively. Therefore, these results further validate the findings of the broth microdilution assay and suggest that the test peptide, at its MFC value, completely inhibits the growth and survival of *C. auris* MRL6057; however, further research will provide an insight into the mode of anti-*Candida* action of the test peptide.

### 3.5. D-lp1 Compromises the Membrane Integrity of C. auris

The PI assay confirmed the membrane disturbed the property of D-lp1 against *C. auris* MRL6057. The obtained data advocated the candidicidal ability of the test plant defensin, which was dose-dependent. The test peptide compromised the integrity of the plasma membrane; therefore, more PI positive *C. auris* MRL6057 cells were visible in the microscopic analysis (Figure 4).

The plasma membrane protects the cellular component, and thus, helps in the sustainability of microbial cells. The antifungal agents targeting the cellular membrane of *C. auris* MRL6057 can, therefore, be a part of anti-*Candida* drug development. However, the PI can only enter cells in a membrane-compromised situation, and the reason for membrane damage could be apoptosis or necrosis. Hence, the cells glowing red during microscopic analysis denote a compromised cell membrane [42]. Thus, the present results advocate that exposure to D-lp1 causes apoptosis or necrosis in *C. auris* MRL6057, which results in disruption of the plasma membrane.

The antimicrobial activity of plant defensin against a wide range of pathogens has been attributed to its ability to interact with fungal and bacterial membrane compounds [18]. Plant defensins such as NaD1 [16,43], MtDef4 [17], and Psd1 [15], upon interaction with fungal cell membranes, become internalized and modulate intracellular targets, while plant defensin, RsAFP2 [44], stay at the outer cellular surface and results in cell death by triggering a crucial signaling cascade. Furthermore, in a previous study, D-lp1 (100–400 µg/mL) was reported to cause membrane damage in *Z. bailii* [23]. Following previous reports, in the current study, D-lp1 was also found to have disruptive membrane activity against the clinical strain of *C. auris*, which triggers cellular death in the *C. auris* MRL6057.

### 3.6. D-lp1 Modulates the Production of SAP Hydrolytic Enzyme

The results demonstrated that exposure to the test plant defensin resulted in the inhibition of extracellular hydrolytic enzymes (SAP) from *C. auris* MRL6057 (Figure 5). The Pz value was calculated, and an increased Pz value reflected decreased SAP production of SAPs. The output of SAP was found unaltered at a lower concentration of the test peptide (0.19 mg/mL; 0.25 × MIC), whereas, with an increasing concentration (0.39 mg/mL; 0.5 × MIC) of D-lp1, the production of extracellular hydrolytic enzymes was lowered by 49.2%. Furthermore, at 0.78 mg/mL (MIC), there was a substantial decrease (78.5%) in enzyme production compared to the unexposed *C. auris* MRL6057.

The *Candida* species is commensal in healthy individuals and an opportunistic pathogen in immunocompromised patients. There is no question that extracellular hydrolytic enzymes are one of the crucial virulence factors and help yeast to establish infection [45]. *C. auris* has been reported to display profound SAP activity; therefore, we investigated the modulatory effect of D-lp1 against SAP production in *C. auris* MRL057. The obtained results suggested that D-lp1 has anti-virulent properties against the hydrolytic activity of proteases in the *C. auris* MRL057 strain. As already mentioned, plant defensins have been widely explored for their antifungal potency against *Candida* species; however, much has not been reported on their anti-virulent property against hydrolytic enzymes in *Candida*. A significant amount of research must be conducted to truly understand the antifungal strength and mode of action of these plant defensins against fungal pathogens. Thus, the present work not only revealed the inhibitory effect of D-lp1 on the growth and viability of *C. auris*, but also unfolded its ability to modulate the hydrolytic activity of SAPs in *C. auris* MRL057.

### 3.7. D-lp1 Impedes the Adherence of C. auris MRL6057 to the Buccal Epithelial Cells

The microscopic examination demonstrated the anti-adherence property of D-lp1 (Figure 6). Compared to the negative control, where the average number of yeast cells attached to the epithelial cells was 430.66, the exposed *C. auris* MRL6057 cells lost the ability to adhere to the epithelial cells. The percentage of inhibition in the attachment of yeast cells to the epithelial cells was 4.72%, 38.62% (* *p* value = 0.0116), and 73.52% (*** *p* value = 0.0007) at 0.19 mg/mL (0.25 × MIC), 0.39 mg/mL (0.5 × MIC), and 0.78 mg/mL (MIC), respectively. Overall, this was a clear indication that D-lp1 has a concentration-dependent effect on the attachment of *C. auris* MRL6057 to buccal epithelial cells.

Adherence of yeast to biotic and abiotic surfaces triggers biofilm formation, and thus, is considered one of the virulence attributes of *Candida* species. The attachment of yeast to host surfaces, such as buccal epithelial cells or denture acrylic surfaces, is a critical parameter for the colonization and development of clinical manifestations of candidiasis [46]. The ability of plant-derived metabolites to impede the adherence of *C. albicans* to human buccal epithelial cells and denture acrylic surfaces has been demonstrated [46], revealing the hidden strength of plant-derived biometabolites against the adherence of yeast to biotic and abiotic surfaces. Similarly, D-lp1, being a plant-derived peptide, showed anti-adherence properties against the attachment of *C. auris* MRL6057 to buccal epithelial cells. This result also compliments the anti-biofilm property of D-lp1 and evokes its candidature for anti-biofilm therapeutics.

### 3.8. D-lp1 Modulates the Functionality of Drug Transporter Efflux Pumps

Drug resistance in *Candida* species is attributed to MDR efflux pumps, and the activity of these transporters can be mapped in vitro with the help of R6G fluorescent dye as a substrate for efflux pumps [47]. The ATP-dependent efflux pumps (ABC superfamily) are responsible for extruding R6G dye in fluconazole-resistant *Candida* species [48]. Therefore, the present study investigated the effect of D-lp1 on efflux pumps of *C. auris* by monitoring the movement of R6G after exposure to the test peptide. In the initial phase of the experiment, the uptake of R6G by *C. auris* MRL6057 (both exposed and unexposed) was reported; after glucose supplementation, the unexposed cells started effluxing R6G, whereas the D-lp1-exposed *C. auris* cells were challenged, and thus, the extent of R6G discharge was very low. The secretion of R6G from the negative control cells clearly showed the presence of active membrane-bound MDR transporter pumps. On the other hand, D-lp1 modulated the activity of these transporters in a concentration-dependent manner. Therefore, the extracellular quantity of R6G remained relatively high upon the addition of glucose (Figure 7).

The modulation of efflux transporters in *C. albicans* by natural products has been previously discussed [33,49]. Additionally, the function of MDR pumps (ABC and MFS transporter) in drug resistance and the genes encoding these transporters in *C. auris* have been investigated [50]. The present investigation agrees with previous reports [51], where more active ABC-type efflux pumps were identified in *C. auris* compared to *C. glabrata*, suggesting their intrinsic resistance to the azole class of antifungals. Additionally, in *Candida* species, antimicrobial peptides have been reported to increase the antifungal activity of conventional drugs by enhancing the anion channel-associated ATP efflux and, therefore, resulting in drug accumulation inside the yeast cells. Thus, the current results demonstrate the modulatory effect of D-lp1 against ABC MDR transporters, similar to that identified in *C. albicans*, which could be helpful in clinical applications.

### 3.9. D-lp1 Impedes Efflux of R6G from C. auris MRL6057

Further confirmation of the modulatory activity of D-lp1 against *C. auris* efflux transporters was performed by an R6G dye accumulation assay. The negative control cells were used to estimate the tendency of R6G accumulation inside *C. auris* cells. The fluorescence microscopy showed dose-dependent activity of the test peptide on R6G accumulation inside cells compared with negative control cells. A high fluorescence was recorded in the cells exposed to 0.78 mg/mL (MIC) of D-lp1 compared to the cells exposed to 0.39 mg/mL (0.5 × MIC) of the test peptide (Figure 8).

The multidrug-resistant property of *C. auris* is mostly attributed to the overexpression of MDR transporters, which in turn enhances the cell’s capability to throw drugs out of the cells and makes them unresponsive to antifungals. The ABC superfamily transporters, such as CaCdr1p and CaCdr2p, are primarily responsible for the development of multidrug resistance in *C. auris* [52,53]. *C. albicans* has been well explored for the activity and inhibition of efflux pumps, whereas fewer details are available on the inhibition of efflux transporters in *C. auris*. The present study demonstrates the efficacy of D-lp1 in blocking efflux transporters in *C. auris* MRL6057.

### 3.10. Hemolytic Activity of D-lp1 against Horse Blood

The cytotoxic activity of D-lp1 is a critical attribute for future antifungal drug developmental analysis. Thus, the test peptide, displaying promising antifungal activity against *C. auris*, was evaluated for its hemolytic activity against horse blood. Triton X was considered to cause 100% hemolysis, while no breakdown was found in the negative control. D-lp1 at a low concentration, 0.19 mg/mL (0.25 × MIC), showed very minimal hemolysis (0.48%); even at other concentrations, 0.39 mg/mL (0.5 × MIC), 0.78 mg/mL (MIC), and 1.56 mg/mL (MFC) the percentage of hemolysis was found to be 1.74%, 9.55%, and 14.72%, respectively (Figure 9). The obtained results agree with previous findings, suggesting the non-toxic behavior of plant defensin against human cells [12,13,14]. Similarly, in a previously published report, D-lp1 did not show any hemolysis against sheep red blood cells, even at the highest concentration (400 µg/mL); D-lp1 lacked any cytotoxic activity against colonic cell lines (Caco-2 cells) [23]. Therefore, these results advocate for the safety of D-lp1 for further in-depth studies and future in vivo investigations against *C. auris* infections.

## 4. Conclusions

The results established candidicidal activity of D-lp1 against twenty-five strains of *C. auris*. Furthermore, D-lp1 was responsible for inhibiting biofilm formation and eradicating mature biofilm in *C. auris*. More importantly, the test peptide abrogated the important virulence attributes (SAP production, adherence) and efficiently modulated/inhibited MDR efflux transporters. It compromised the cell membrane integrity in *C. auris* MRL6057. Additionally, D-lp1 displayed low hemolytic activity against horse blood, and thus, advocated its use to check infectious diseases caused by this emerging pathogenic yeast, *C. auris*. Further clinical studies are required to confirm the rationality of these results.

## Figures and Tables

**Figure 1 jof-08-01298-f001:**
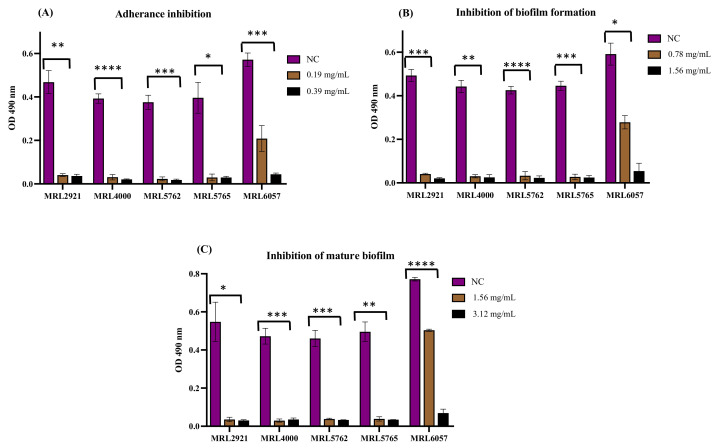
Antibiofilm activity of D-lp1 against *C. auris* biofilms. (**A**) Adherance inhibition; (**B**) Inhibition of biofilm; (**C**) Inhibition of mature biofilm. The figure shows the anti-biofilm potency of the test agent against different stages of biofilms formed by clinical strains of *C. auris*. XTT assay calculated the metabolic activity of cells embedded in biofilms, and readings were recorded at 490 nm. NC, negative control. Statistical difference between test concentrations relative to the negative control was evaluated using a two-way ANOVA, with post hoc Dunnet’s test for multiple comparisons **** *p* value < 0.0001; *** *p* value = 0.0004; ** *p* value = 0.0027; * *p* value = 0.0121.

**Figure 2 jof-08-01298-f002:**
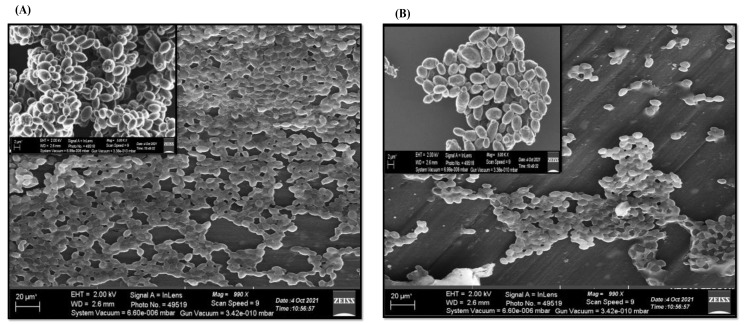
SEM images of *C. auris* MRL6057. The figure demonstrates the disruptive ability of D-lp1 against 24-h mature biofilm formed by *C. auris* MRL6057. (**A**) represents untreated control, (**B**) represents abrogated *C. auris* biofilm after treatment with D-lp1 at 3.12 mg/mL. The image at higher magnification has been placed in the upper left corner.

**Figure 3 jof-08-01298-f003:**
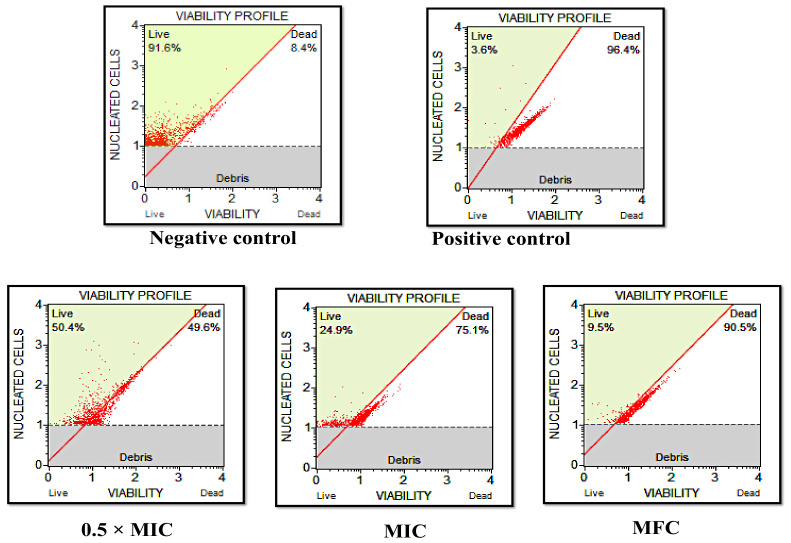
D-lp1 affects *C. auris* MRL6057 cell viability. The figure shows *C. auris* viability profile. Negative control: unexposed *C. auris* cells; positive control: H_2_O_2_ exposed cells; *C. auris* exposed to varied MIC values of the test peptide; 0.39 mg/mL (0.5 × MIC); 0.78 mg/mL (MIC) of 1.56 mg/mL (MFC).

**Figure 4 jof-08-01298-f004:**
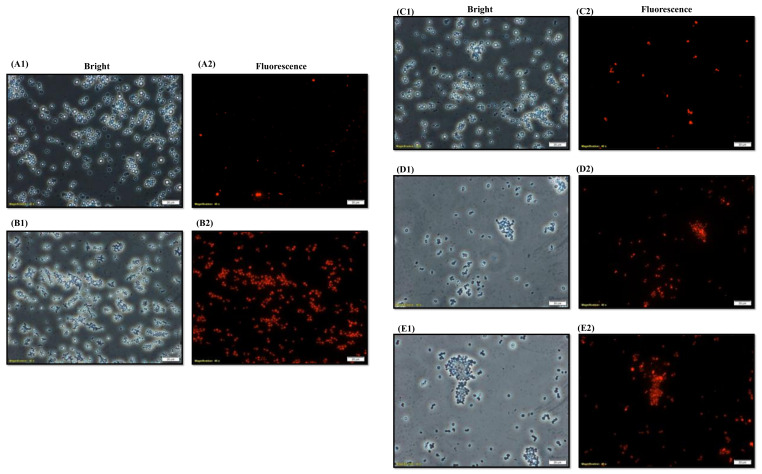
Uptake of PI by *C. auris* MRL6057. Exposure of yeast cells by D-lp1 at various concentrations (**C**–**E**). (**A**) Untreated cells were a negative control for intact *C. auris* plasma membrane. (**B**) Exposure to H_2_O_2_ caused compromised cell membrane resulting in cellular uptake of PI. Exposure of *C. auris* at (**C**) 0.39 mg/mL (0.5 × MIC); (**D**) 0.78 mg/mL (MIC); (**E**) 1.56 mg/mL (MFC).

**Figure 5 jof-08-01298-f005:**
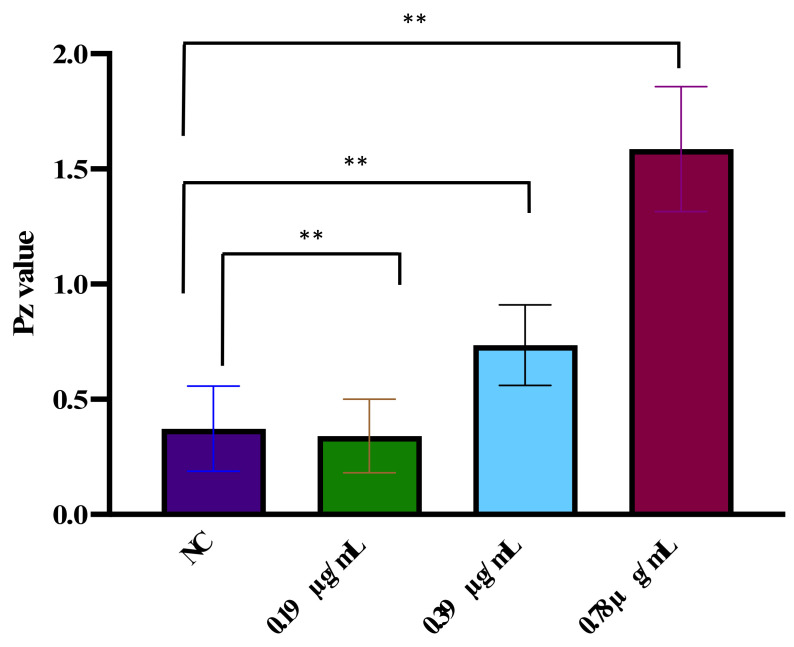
Inhibition of SAP production in *C. auris* MRL6057 by the D-lp1. The figure represents the average Pz value (diameter of colony/total diameter) in exposed and unexposed (NC) cells. 0.19 mg/mL, 0.25 × MIC; 0.39 mg/mL, 0.5 × MIC; 0.78 mg/mL, MIC. Student’s unpaired two-tailed *t*-tests assuming unequal variance, ** *p* value = 0.003.

**Figure 6 jof-08-01298-f006:**
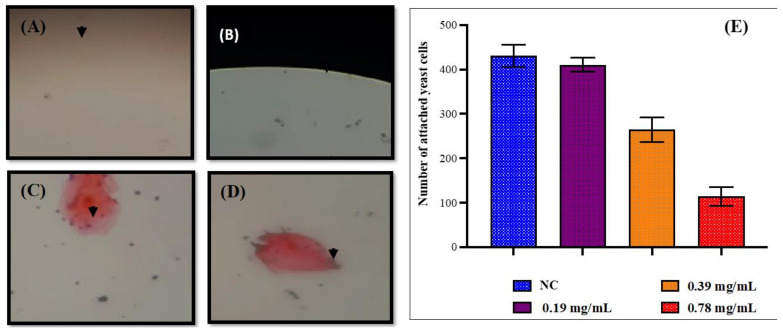
Adherence of *C. auris* MRL6057 to buccal epithelial cells. (**A**) Negative control shows the attachment of unexposed *C. auris* to the epithelial cells. The anti-adherence property of D-lp1 at various concentrations, (**B**) 0.19 mg/mL, 0.25 × MIC; (**C**) 0.39 mg/mL, 0.5 × MIC; (**D**) 0.78 mg/mL, MIC. The arrow points out the attached yeast cells to the epithelial cells. (**E**) represents the number of yeast cells attached to the buccal epithelial cells in the negative control (NC) and at various concentrations of D-lp1.

**Figure 7 jof-08-01298-f007:**
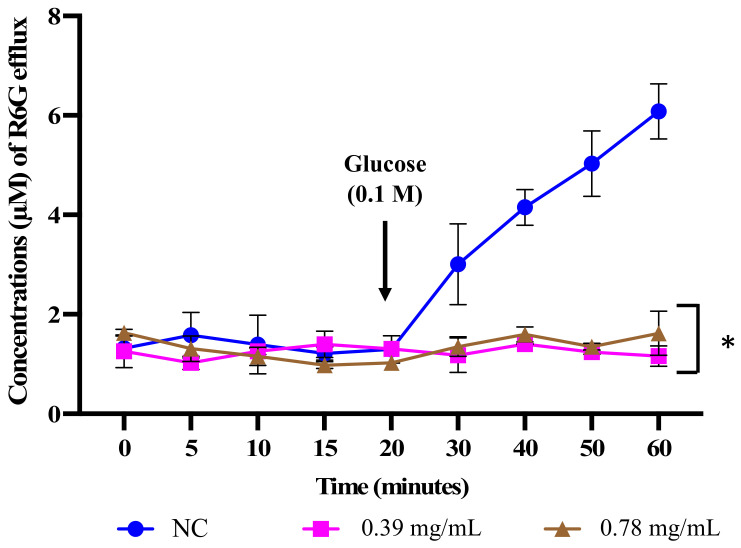
D-lp1 modulates extracellular efflux of R6G. The image displays the extracellular quantity of R6G effluxed by *C. auris* MRL6057. The absorbance of extracellular R6G was recorded at 527 nm. NC, negative control. The values are the average of three independent experiments. 0.39 mg/mL, 0.5 × MIC; 0.78 mg/mL, MIC. Student’s unpaired two-tailed *t*-tests assuming unequal variance, * *p* value < 0.05.

**Figure 8 jof-08-01298-f008:**
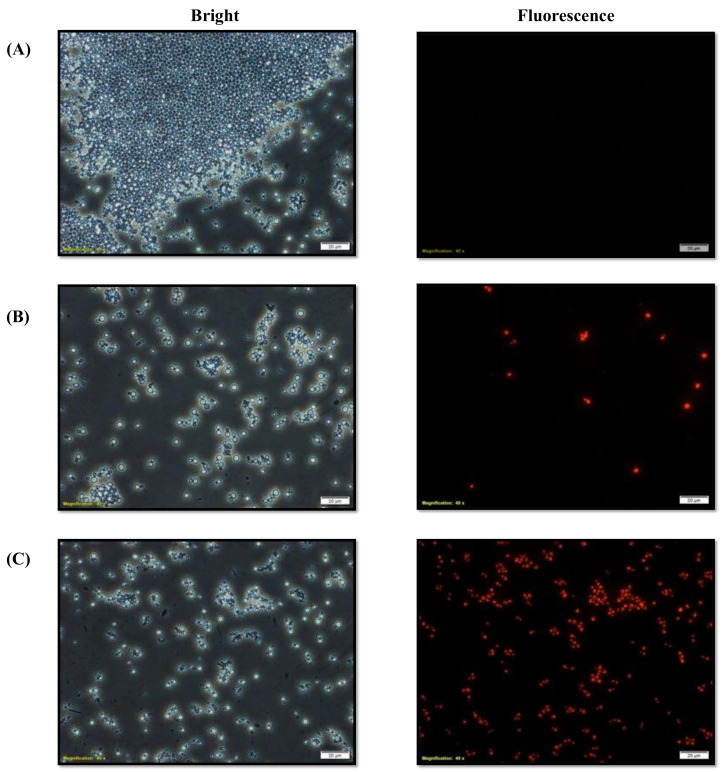
Intracellular accumulation of R6G in *C. auris*. In the figure, *C. auris* MRL6057 was observed under bright field and fluorescence microscopy. (**A**) The figure shows untreated *C. auris* cells with effluxed R6G dye during incubation with glucose. Whereas treatment with 0.39 mg/mL, 0.5 × MIC (**B**), and 0.78 mg/mL MIC (**C**) resulted in the accumulation of R6G inside the cells after incubation with glucose, which was represented by high fluorescence inside the cells.

**Figure 9 jof-08-01298-f009:**
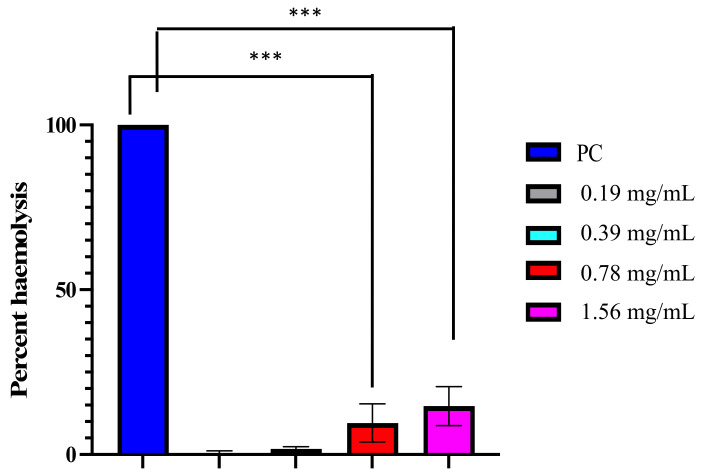
Hemolytic activity of D-lp1 against horse blood. Positive control (PC) 1% Triton X-100, Negative control (NC). Hemolytic activity of D-lp1 at varied concentrations; *** *p* value < 0.0004.

**Table 1 jof-08-01298-t001:** Minimum inhibitory concentrations (MIC) and minimum fungicidal concentrations (MFC) for D-lp1 against different clinical strains of *Candida auris*.

Groups	*C. auris*	MIC (mg/mL)	MFC (mg/mL)	Groups	*C. auris*	MIC (mg/mL)	MFC (mg/mL)
**A**	MRL3785 MRL6125 MRL6173 MRL6183 MRL6194 MRL6326 MRL6015 MRL6338	0.047	0.095	**B**	MRL3499 MRL4587 MRL5418 MRL6059 MRL6277 MRL6333 MRL6334 MRL6339 MRL2397 MRL4888 MRL6005 MRL6065	0.095	0.19
**C**	MRL2921 MRL4000 MRL5762 MRL5765 MRL6057	0.78	0.156				
*C. albicans* ATCC90028		0.047	0.188				
*C. parapsilosis* ATCC22019		0.095	0.376				

## Data Availability

Not applicable.

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
