# Peer review of "Characterization of Defensin-like Protein 1 for Its Anti-Biofilm and Anti-Virulence Properties for the Development of Novel Antifungal Drug against Candida auris"

_jof, 2022, doi:10.3390/jof8121298_

Round 1

Reviewer 1 Report

The manuscript entitled “Characterization of defensin-like protein 1 for its anti-biofilm and anti-virulence properties for the development of novel antifungal drug against Candida auris” by Kamli et al., reports the anticandidal activity of D-lp1 against multidrug resistant C. auris. While the paper is timely important and the objective of the work is also important There are several refinements required before the paper can be considered for publication. Below are some of the comments which need to be addressed.

1.      From the SEM images (fig 2) it is evident that the biofilm is disrupted by D-lp1 but the cells remained intact. Can you suggest which mechanism might be responsible for the antibiofilm activity of D-lp1.

2.      The activity of D-1p1 on SAP production has been detailed; however nothing has been mentioned if the effect is at the protein level or related to its genes. It would have been ideal to include molecular studies of gene expression related to these genes.   

3.      Authors used 1% DMSO as a solvent to solubilize the peptide and drugs. Whereas in methods it is mentioned that 1% DMSO is used as control; nothing has been mentioned in the results. Please highlight if DMSO has affected the growth or any other parameter studied.

Some Minor comments are mentioned below:

In lines # 19 to 20 the sentence "Furthermore, the D-lp1 was 20 analyzed for its anti-biofilm and anti-virulence properties following standard protocols" shoulde be replaced by "Furthermore, following standard protocols, the D-lp1 was analyzed for its anti-biofilm and anti-virulence properties".

 in the Line 86       kept in a dark should be replaced by kept in the dark

in Line 177 "4 h time" should be replaced by "4h".

In Line 252 "was found active" should be replaced by "was found to be active".

In Line 353 'the PI can only enter cells with" should be replaced by the "PI can only enter cells in"

In Line 361 "modulates" should be replaced by "modulate".

Line 433 the sentence "Therefore, the extracellular quantity of R6G did not increase much upon the addition of glucose (Figure 7)" Should be replaced by Therefore, the extracellular quantity of R6G remained relatively high upon the addition of glucose (Figure 7).

Line 453  done by performing should be replaced by performed by

Author Response

Comments and Suggestions for Authors

The manuscript entitled “Characterization of defensin-like protein 1 for its anti-biofilm and anti-virulence properties for the development of novel antifungal drug against Candida auris” by Kamli et al., reports the anticandidal activity of D-lp1 against multidrug resistant C. auris. While the paper is timely important and the objective of the work is also important There are several refinements required before the paper can be considered for publication. Below are some of the comments which need to be addressed.

Comment 1: From the SEM images (fig 2) it is evident that the biofilm is disrupted by D-lp1 but the cells remained intact. Can you suggest which mechanism might be responsible for the antibiofilm activity of D-lp1.

Response 1: Figure-2 shows the anti-biofilm property of the test peptide and the probable mode of anti-biofilm action could be the ability of D-lp1 to modulate quorum sensing as well as downregulating crucial genes associated with the biofilms. Since the present study was a pilot study but it gave us important stepping information which will be used for further in-depth investigation and exploring the candidicidal activity of D-lp1.

Comment 2: The activity of D-1p1 on SAP production has been detailed; however nothing has been mentioned if the effect is at the protein level or related to its genes. It would have been ideal to include molecular studies of gene expression related to these genes.   

Response 2: Authors are interested to work on the SAPs virulence-related gene expression; however, C. auris is not well explored and has varying reports of virulence factors in different clades. C. auris encodes orthologs of many known C. albicans lytic enzymes, including secreted aspartyl proteases, secreted lipases, and phospholipases. Several studies have demonstrated lytic activity for C. auris SAPs (doi: 10.1038/s41426-018-0187-x) but little is known about how the expression profiles of these enzymes compared to their C. albicans orthologs (doi: 10.3389/fgene.2020.00351). Further molecular and virulence studies are necessary to determine the expression profile of SAPs and their role in C. auris infection strategy. It would’ve been interesting to see these results in this study; however, as mentioned this is a pilot study and such studies would have deviated from the central thesis of this study.

Comment 3: Authors used 1% DMSO as a solvent to solubilize the peptide and drugs. Whereas in methods it is mentioned that 1% DMSO is used as control; nothing has been mentioned in the results. Please highlight if DMSO has affected the growth or any other parameter studied.

Response 2: Thank you for raising this point. The results demonstrated that there was no impact of DMSO in the growth of C. auris. The observation has been added to the revised manuscript.

# Some Minor comments are mentioned below:

  • In lines # 19 to 20 the sentence "Furthermore, the D-lp1 was 20 analyzed for its anti-biofilm and anti-virulence properties following standard protocols" shoulde be replaced by "Furthermore, following standard protocols, the D-lp1 was analyzed for its anti-biofilm and anti-virulence properties".
  • in the Line 86, kept in a dark should be replaced by kept in the dark
  • in Line 177 "4 h time" should be replaced by "4h".
  • In Line 252 "was found active" should be replaced by "was found to be active".
  • In Line 353 'the PI can only enter cells with" should be replaced by the "PI can only enter cells in"
  • In Line 361 "modulates" should be replaced by "modulate".
  • Line 433 the sentence "Therefore, the extracellular quantity of R6G did not increase much upon the addition of glucose (Figure 7)" Should be replaced by Therefore, the extracellular quantity of R6G remained relatively high upon the addition of glucose (Figure 7).
  • Line 453 done by performing should be replaced by performed by

Response: Thank you for bringing this to our notice, all suggestions have been incorporated into the revised manuscript.

Reviewer 2 Report

This is an interesting manuscript that describes the evaluation of a plant defensin against the pathogen Candida auris.  This resistant organism poses an important problem, so the work is a significant contribution to potential approaches to treatment.  The work is thorough, with a number of different experiments being used to define the activity and to suggest a mechanism of action.  In addition, some experiments indicate that this compound may have low cytotoxicity.  The experiments are clearly described and the results well presented with high quality graphics.  Collectively, the paper reveals promise for this approach.

That said, there are small but significant grammatical issues throughout, which should be fixed prior to final publication. 

One other scientific point is that the units of concentration for defensins are generally given in mg/mL, whereas some positive control antibiotics e.g. amphotericin B are in micrograms/mL (see e.g. line 95).  This doesn't help the reader gain a sense of the relative activity, which is lower for the defensin.  While this lower activity may seem problematic, it may not be given the acute toxicity of amphotericin B, compared to the apparent lack of cytotoxicity of defensin.  This could be pointed out in the text.

I did wonder about the potential for proteases (or peptidases, etc.) to hydrolyze the defensins, which could be problematic.  This point was not made, but perhaps this is beyond the scope of the present work.

Overall, once these points have been addressed, this manuscript is interesting and merits publication.

Some small points:

Abstract, line 3: "more severe"  more than what?

Line 37: "it has built up...."  what is "it"?  Advancements from the previous sentence?  If so then they are plural, so "they" not "it"

Although generally correct, please check throughout: spaces between numbers and units, except for degrees, e.g. 1 g but 1oC.  See e.g. lines 167-169

Author Response

 Comments and Suggestions for Authors

This is an interesting manuscript that describes the evaluation of a plant defensin against the pathogen Candida auris.  This resistant organism poses an important problem, so the work is a significant contribution to potential approaches to treatment.  The work is thorough, with a number of different experiments being used to define the activity and to suggest a mechanism of action.  In addition, some experiments indicate that this compound may have low cytotoxicity.  The experiments are clearly described and the results well presented with high quality graphics.  Collectively, the paper reveals promise for this approach.

Comment 1: That said, there are small but significant grammatical issues throughout, which should be fixed prior to final publication. 

Response 1: As suggested, the entire manuscript was thoroughly checked and all the grammatical issues were corrected in the revised manuscript.

Comment 2: One other scientific point is that the units of concentration for defensins are generally given in mg/mL, whereas some positive control antibiotics e.g. amphotericin B are in micrograms/mL (see e.g. line 95). This doesn't help the reader gain a sense of the relative activity, which is lower for the defensin. While this lower activity may seem problematic, it may not be given the acute toxicity of amphotericin B, compared to the apparent lack of cytotoxicity of defensin.  This could be pointed out in the text.

Response 2: The authors agree with your point of view that variation in the units of control drug and test peptide may create a sense of confusion among readers, therefore, the concentration units have been made consistent throughout the manuscript. Also, the non-cytotoxic nature D-lp1 has been discussed in terms of AmB in the revised text.

Comment 3: I did wonder about the potential for proteases (or peptidases, etc.) to hydrolyze the defensins, which could be problematic.  This point was not made, but perhaps this is beyond the scope of the present work.

Response 3: The authors agree that hydrolyzation of defensin could be possible, but this is a matter of further in depth investigation. Since this was a pilot study and therefore, this aspect falls beyond the scope of the present work. 

Comment 4: Overall, once these points have been addressed, this manuscript is interesting and merits publication. Some small points:

Abstract, line 3: "more severe"  more than what?

Line 37: "it has built up...."  what is "it"?  Advancements from the previous sentence?  If so then they are plural, so "they" not "it"

Although generally correct, please check throughout: spaces between numbers and units, except for degrees, e.g. 1 g but 1oC.  See e.g. lines 167-169

Response 4: As suggested, all the suggestions stated above have been incorporated in the revised version for a better understanding of our work.

Reviewer 3 Report

The author did a careful study on the utility of defensin-like protein1 on treatment of Candida strains. Please see the comments below:

  1. Please add description of the structure and function knowledge of defensin like protein 1 in the introduction.

  2. For table 1, please list individual MIC, MFC values for each strain, if they are from previously published reports, please cite the original reports.

  3. Please re-plot figure 5. You can change it to bar plot, scatter plot is not needed in this case since you don’t have value for x axis.

Author Response

Comments and Suggestions for Authors

The author did a careful study on the utility of defensin-like protein1 on treatment of Candida strains. Please see the comments below:

Comment 1: Please add description of the structure and function knowledge of defensin like protein 1 in the introduction.

Response 1: Thank you for your suggestion. Details about D-lp1 have been appended in the revised manuscript.

Comment 2: For table 1, please list individual MIC, MFC values for each strain, if they are from previously published reports, please cite the original reports.

Response 2: Thank you for your comment. We used 25 clinical strains of Candida auris, they were tested against D-lp1 and were grouped based on the MIC and MFC values. The purpose of keeping one MIC, MFC value for each group was to minimize the repetition of values and to enhance the understanding of readers. Moreover, it gives a concise overview of our findings. Keeping individual values for each strain will not add any further information about their antifungal susceptibility profiling.

Furthermore, we have cited previous publications to show the antifungal susceptibility profiling of these strains against standard antifungal drugs; the activity of D-lp1 against these clinical isolates has not been published previously.       

Comment 3: Please re-plot figure 5. You can change it to bar plot, scatter plot is not needed in this case since you don’t have value for x axis.

Response 3: As suggested, we have changed the scatter plot to a bar graph for better understanding in the new submission.